

# Nonlinear modes disentangle glassy and Goldstone modes in structural glasses

## L. Gartner and E. Lerner

Institute for Theoretical Physics, University of Amsterdam,Science Park 904, 1098 XH Amsterdam, The Netherlands

## Abstract

One outstanding problem in the physics of glassy solids is understanding the statistics and properties of low-energy excitations that stem from the disorder that characterizes these systems' microstructure. In this work we introduce a family of algebraic equations whose solutions represent collective displacement directions (modes) in the multidimensional configuration space of a structural glass. We explain why solutions of the algebraic equations, coined nonlinear glassy modes, are quasi-localized low-energy excitations. We present an iterative method to solve the algebraic equations, and use it to study the energetic and structural properties of a selected subset of their solutions constructed by starting from a normal mode analysis of the potential energy of a model glass. Our key result is that the structure and energies associated with harmonic glassy vibrational modes and their nonlinear counterparts converge in the limit of very low frequencies. As nonlinear modes never suffer hybridizations, our result implies that the presented theoretical framework constitutes a robust alternative definition of 'soft glassy modes' in the thermodynamic limit, in which Goldstone modes overwhelm and destroy the identity of low-frequency harmonic glassy modes.

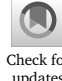

# 1 Introduction

Low-frequency vibrational modes in structural glasses stem from two very different physical origins: the translational symmetry of the Hamiltonian gives rise to Goldstone modes, whose frequencies are known to be distributed according to the Debye law $\omega^{\bar{d}-1}$, where $\bar{d}$ denotes the spatial dimension and $\omega$ the frequency. In addition to the Goldstone modes, the disordered nature of the glass microstructure gives rise to a population of soft vibrational modes (at least in small enough systems, see below), which are believed to be central players in determining many thermodynamic [1–5], dynamic [6,7], and solid-mechanical [8–11] glassy phenomena. While the occurrence of Goldstone modes is well-understood, the situation is far less clear for soft glassy modes, whose precise origin [12–18], structure [19–21], and implications [22,23] have been the focus of much attention for several decades now.

In a solid of linear size $L$, the frequency of the lowest Goldstone mode is estimated as $2\pi\sqrt{\mu/\rho}/L$, where $\mu$ is the athermal shear modulus, $\rho \equiv mN/L^{\bar{d}}$ is the density, and $N$ the number of particles of mass $m$ [24]. In a recent computational study [21] this relation between the lowest Goldstone mode frequency and system size was exploited: decreasing the size of glassy samples in three dimensions (3D) pushed the frequencies of Goldstone modes up, cleanly exposing a population of low-frequency glassy modes. It was shown in several popular glass forming models that in systems small enough to sufficiently suppress low-frequency Goldstone modes, the density of vibrational frequencies (also referred to as the density of states (DOS)) grows from zero as $D(\omega) \sim \omega^4$ up to the vicinity of the lowest Goldstone mode frequency. In what follows, we refer to the vibrational (normal) modes that populate the frequency regime in which the $\omega^4$ law holds as *harmonic glassy modes* (HGMs). In [21] it was further shown that HGMs are quasilocalized; they display a disordered core decorated by 'continuum-like' fields that decay at distances $r$ away from the core as $r^{-2}$ in 3D.

In this work we study the intrusion of Goldstone modes into the glassy modes' frequency regime by systematically increasing the system size. We show (see Fig. 1 below) how strong hybridizations then occur that severely destroy the quasi-localized nature of harmonic glassy modes. This demonstration rules out the possibility that quasi-localized soft glassy modes can be represented at all as harmonic normal modes in large systems, and in particular in the thermodynamic limit, in which Goldstone modes dominate the low-frequency regime. This raises a crucial question: is there a system-size-independent way to define these glassy quasilocalized excitations, overcoming the destruction of their identities as normal modes by hybridization with Goldstone modes?

One approach to overcome the issue of hybridization and mixing of glassy modes with Goldstone modes in large systems involves the introduction of auxiliary terms to the Hamiltonian that result in the suppression of Goldstone modes at low frequencies. Soft glassy modes

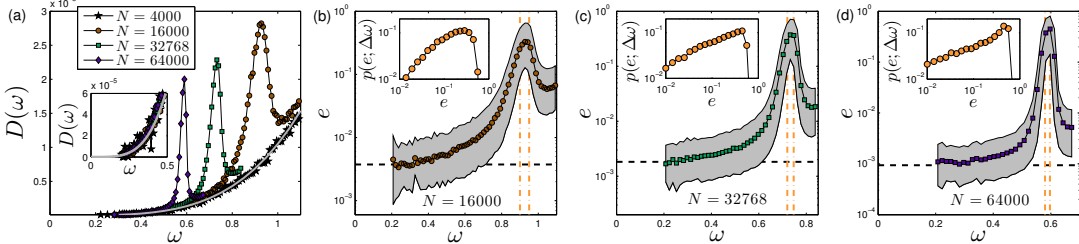

Figure 1: (a) Density of states $D(\omega)$ for various system sizes (as indicated by the legend), plotted up to frequencies slightly larger than the lowest Goldstone mode frequency. The continuous line represents the $D(\omega) \sim \omega^4$ law. The inset shows the same data up to $\omega = 0.5$, see text for further discussion. (b)-(d) Participation ratio $e$ vs. frequency; the symbols represent the means, binned over frequency. For each bin, we also calculated 10 deciles; the 2nd-9th deciles are represented by the shaded area, which covers 80% of the data. The dashed horizontal lines represent the scaling $e = c/N$ with $c \approx 60$. The insets show the distribution of participation ratios, for modes with frequencies that fall in the range $\Delta\omega$ whose limits are marked by the vertical dash-dotted lines. These data show clearly that due to hybridizations, glassy modes are very unlikely to maintain their localized character if their frequencies fall in the vicinity of Goldstone modes' frequencies.

are then manifested as normal modes of the (harmomic approximation of the) modified Hamiltonian. Wijtmans and Manning took such an approach in [25], where it was shown that the additional terms in the Hamiltonian also leads to a suppression of the continuum-like far fields that decorate the core of soft glassy modes. The glassy modes, termed 'defects' in [25], were shown to decay exponentially in space, leading to an increase in their energies. A similar approach was taken by the authors of [26] in the study of low-energy excitations in the Heisenberg spin glass in 3D. There, a random magnetic field was added to the spin-glass Hamiltonian, pushing Goldstone modes (in this case, spin-waves) to higher frequencies, and exposing a population of glassy modes, also found to follow a $\omega^4$ law. In [27] the effects of pinning particles on the density of states in a 2D system of hard discs was studied, while the focus there was on the properties of systems approaching the jamming transition.

In this work we take an orthogonal approach to the efforts reviewed above; instead of modifying the Hamiltonian such that soft glassy modes become represented by normal modes, we rather introduce a theoretical framework which embeds a definition of a family of soft quasi-localized nonlinear excitations that are entirely indifferent to the proximity of low-lying Goldstone modes. We show that the spatial structure of these excitations shares the same features as found for harmonic glassy modes: they are also centered on a disordered core decorated by long-range continuum-like fields. We further show that their associated quadratic energies converge to those of harmonic modes in the limit of small frequencies. These excitations are therefore very good representatives of harmonic glassy modes that would exist in the thermodynamic limit if Goldstone modes could be completely suppressed.

This paper is organized as follows. In the next Section we breifly review the numerical models and methods used in this study; further details are provided in Appendix A. In Sect. 3 we present data that demonstrates the intrusion of Goldstone modes into the glassy modes' regime, and show signatures of the strong hybridization of harmonic glassy modes with Goldstone modes. In Sect. 4 a set of algebraic equations is introduced whose solutions are coined *nonlinear glassy modes* (NGMs), and the nature of these solutions is discussed. In Sect. 5 a mapping is introduced that can be use in an iterative scheme for finding solutions to the algebraic equations. Here we further demonstrate that extended modes are always mapped to quasi-localized modes by the iterative scheme. In Sect. 6 we study various structural and en-

ergetic properties of nonlinear modes, and present our key result discussed above. In Sect. 7 we directly demonstrate how nonlinear modes disentangle glassy and Goldstone modes. We summarize our findings and discuss future work in Sect. 8.

## 2 Methods and models

We employed a three dimensional 50:50 binary mixture of 'small' and 'large' point-like particles interacting via inverse power-law purely repulsive pairwise potentials, under fixed volume with periodic boundary conditions. A few issues are demonstrated visually in what follows using two-dimensional systems, however all reported data is shown for 3D systems. We study systems with sizes ranging between $N = 10^3$ to $N = 10^6$. Samples were prepared by a quick quench from the equilibrium liquid phase. Decay profiles of various fields were calculated as described in Refs. [11, 21, 28]. Nonlinear modes were calculated using the iterative method introduced in Sect. 5. For a detailed description of our models and numerical methods, see Appendix A.

## 3 Hybridization of glassy and Goldstone modes

Here we demonstrate the intrusion of Goldstone modes into the harmonic glassy modes' frequency regime in the density of vibration modes by gradually increasing the system size until the suppression of Goldstone modes is lifted. In Fig. 1a the DOS of our model glass is plotted for different system sizes; the first Goldstone modes are apparent in the form of peaks that shift to the left as $N$ is increased. We also plot the DOS for systems of $N = 4000$ for reference, and superimpose the $\omega^4$ law (continuous line). In samples of our model system prepared by a quick quench from the liquid phase, the Debye frequency $\omega_D \approx 17$ (in the appropriate microscopic units, see Appendix A for further details), which means that the glassy modes' regime is found for frequencies $\omega/\omega_D \lesssim 0.05$, conditioned that Goldstone modes do not dwell there. Interestingly, since we perform an ensemble average of the DOS over at least 5000 independent configurations, the $\omega^4$ law is found to hold below the lowest Goldstone frequency in all system sizes presented (see inset of Fig. 1a), despite that in some realizations (and more so in larger systems) the lowest frequency mode is a Goldstone mode [21].

How does the intrusion of Goldstone modes into the glassy modes' regime effect their localization properties? We approach this question by focusing on the participation ratio

$$e \equiv \frac{1}{N \sum_i (\hat{\Psi}_i \cdot \hat{\Psi}_i)^2} \, , \tag{1}$$

which is a simple measure of the degree of localization of a mode; fully localized modes would have $e \sim 1/N$, whereas delocalized modes have participation ratios on the order of unity. In panels (b)-(d) of Fig. 1 we plot the participation ratio vs. frequency; the symbols represent the means, binned over frequency, while the shaded areas exclude 10% of the lowest and highest participation ratios, per frequency bin (i.e. it covers 80% of the data points). This analysis indicates that in all system sizes the modes that populate asymptotically low frequencies are localized modes, as can be seen by comparing the participation ratio data to the horizontal dashed lines, which follow a $1/N$ scaling. Importantly, it also shows that it is extremely unlikely to find localized modes with frequencies in the vicinity of Goldstone modes' frequencies, due to strong hybridization effects.

In the next Section we introduce nonlinear glassy modes which are shown to be quasi-localized, similarly to harmonic glassy modes. However, nonlinear glassy modes do not suffer

hybridizations, even when their energies are comparable to Goldstone modes' energies.

## 4 Definition of nonlinear glassy modes

In this Section we introduce our definition of nonlinear glassy modes, and explain in detail why they are both quasilocalized, and have low energies.

Nonlinear glassy modes (NGMs) of order $n$ are defined as solutions $\hat{\pi}$ of the following equation

$$\mathcal{M} \cdot \hat{\pi} = \frac{\mathcal{M} : \hat{\pi}\hat{\pi}}{U^{(n)} \bullet \hat{\pi}^{(n)}} U^{(n)} \bullet \hat{\pi}^{(n-1)} \,, \tag{2}$$

where $\mathcal{M} \equiv \frac{\partial^2 U}{\partial \vec{x} \partial \vec{x}}$ denotes the dynamical matrix, and $\vec{x}$ are the particles' coordinates. The notation $U^{(n)}$ stands for the rank-$n$ tensor of derivatives of the potential energy $U$, and the combination $\bullet \hat{\pi}^{(n)}$ denotes a contraction over $n$ instances of the vector $\hat{\pi}$. For example, for $n = 3$ and $n = 4$ Eq. (2) reads

$$\mathcal{M}_{ij} \cdot \hat{\pi}_j = \frac{\mathcal{M}_{qj} : \hat{\pi}_q \hat{\pi}_j}{\frac{\partial^3 U}{\partial \vec{x}_q \partial \vec{x}_j \partial \vec{x}_k} \vdots \hat{\pi}_q \hat{\pi}_j \hat{\pi}_k} \frac{\partial^3 U}{\partial \vec{x}_i \partial \vec{x}_j \partial \vec{x}_k} : \hat{\pi}_j \hat{\pi}_k \,, \tag{3}$$

and

$$\mathcal{M}_{ij} \cdot \hat{\pi}_j = \frac{\mathcal{M}_{qj} : \hat{\pi}_q \hat{\pi}_j}{\frac{\partial^4 U}{\partial \vec{x}_q \partial \vec{x}_j \partial \vec{x}_k \partial \vec{x}_\ell} \vdots \hat{\pi}_q \hat{\pi}_j \hat{\pi}_k \hat{\pi}_\ell} \frac{\partial^4 U}{\partial \vec{x}_i \partial \vec{x}_j \partial \vec{x}_k \partial \vec{x}_\ell} \vdots \hat{\pi}_j \hat{\pi}_k \hat{\pi}_\ell \,, \tag{4}$$

respectively. Here and in what follows, repeated subscript indices, that label particles, are assumed to be summed over, unless explicitly indicated otherwise. Wherever unnecessary, we omit the particle indices, then e.g. $\hat{\pi}, \hat{\Psi}$, or $\hat{z}$ denote a normalized $N \times \bar{d}$ dimensional displacement direction in the configuration space of our system.

The $n = 3$ case, as spelled out in Eq. (3), was proposed before in studies of plastic instabilities in athermally deformed disordered solids [11, 28]. In those studies, solutions $\hat{\pi}$ were coined *nonlinear plastic modes* due to their particular relevance to micromechanical processes of plastic instabilities. It was shown that for $n = 3$ solutions $\hat{\pi}$ pertain to minima of a 'barrier function' $b(\hat{z})$, defined as

$$b(\hat{z}) \equiv \frac{2}{3} \frac{\left( \mathcal{M}_{ij} : \hat{z}_i \hat{z}_j \right)^3}{\left( \frac{\partial^3 U}{\partial \vec{x}_i \partial \vec{x}_j \partial \vec{x}_k} \vdots \hat{z}_i \hat{z}_j \hat{z}_k \right)^2} \,. \tag{5}$$

The function $b(\hat{z})$ represents the height of the potential energy barrier that separates neighboring inherent states, assuming a cubic expansion of the potential energy variation stemming from the displacement of particles along a general direction $\hat{z}$ [11, 28]. A straightforward generalization of this idea can be spelled out for nonlinear glassy modes of any order. To this aim, we define the $n$th order *cost functions*

$$\mathcal{G}_n(\hat{z}) \equiv \frac{(\mathcal{M} : \hat{z}\hat{z})^n}{(U^{(n)} \bullet \hat{z}^{(n)})^2} \,. \tag{6}$$

We note that $\mathcal{G}_n(\hat{z})$ has units of energy$^{n-2}$, and for orders $n > 3$ it lacks a clear physical interpretation. Nevertheless, it is easily shown that solutions $\hat{\pi}$ to Eq. (2) pertain to stationary points, and therefore in particular to *local minima*, of $\mathcal{G}_n(\hat{z})$. This means that such solutions can be easily found numerically by nonlinear minimization techniques of the appropriate cost function, as demonstrated in [11, 28] for $n = 3$.

The pertaining of NGMs to minima of the cost function $\mathcal{G}_n(\hat{z})$ leads to two important conclusions: first, since the stiffness of a NGM $\kappa \equiv \mathcal{M} : \hat{\pi}\hat{\pi}$ appears in the numerator of $\mathcal{G}_n(\hat{z})$, we

conclude that NGMs are directions $\hat{\pi}$ associated with *small* stiffnesses. This directly connects between NGMs and low-frequency vibrational modes $\hat{\Psi}$, whose stiffnesses are represented by their associated eigenvalues $\lambda = \mathcal{M} : \hat{\Psi}\hat{\Psi} = \omega^2$ (recalling that all masses $m = 1$).

Next, notice that $U^{(n)} \bullet \hat{\pi}^{(n)}$ are Taylor expansion coefficients of the energy variation $\delta U(s) \equiv U(s) - U(0)$ upon displacing the particles a distance $s$ along $\hat{\pi}$, namely

$$\delta U(s) = \sum_{n>1} \frac{U^{(n)} \bullet \hat{\pi}^{(n)}}{n!} s^n . \tag{7}$$

The second conclusion from the discussion above is that NGMs are associated with *large* expansion coefficients $U^{(n)} \bullet \hat{\pi}^{(n)}$, since the latter appear in the denominator of the cost function $\mathcal{G}_n(\hat{z})$, which is minimized by $\hat{\pi}$.

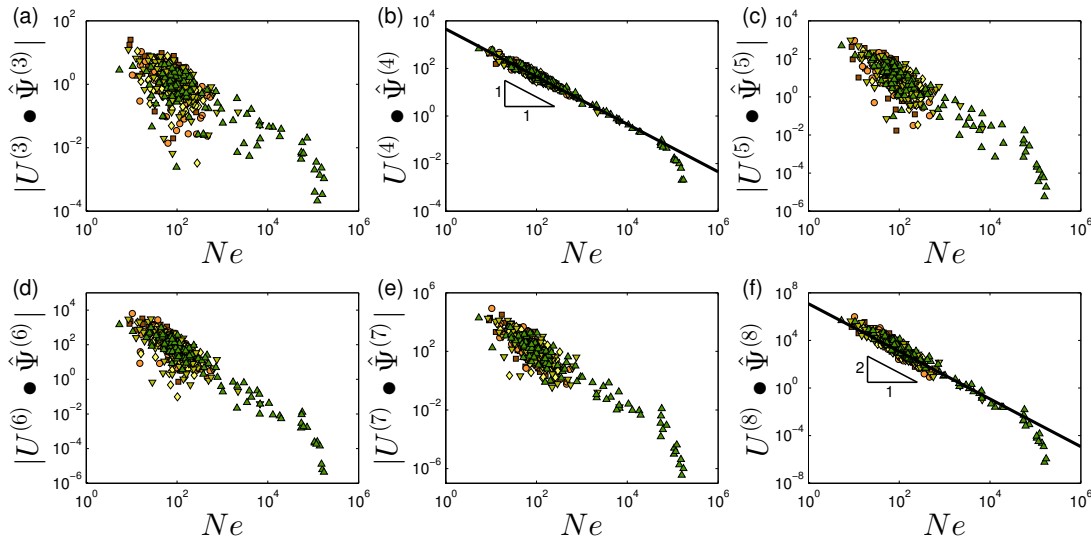

Figure 2: Expansion coefficients (see text for definition) vs. $N$ times the participation ratio $e$ of normal modes $\hat{\Psi}$. We show the absolute values for $n = 3, 5, 6, 7$ and the bare numbers for $n = 4, 8$ which are always found to be positive in our model. Data is measured for the lowest-frequency mode of 100 independent samples for each system size, with $\square$, $\circ$, $\lozenge$, $\triangledown$ and $\triangle$ representing systems of $N = 1000, 4000, 16000, 64000$, and $256000$, respectively. A clear general trend appears: the magnitude of expansion coefficients is larger the more localized a mode is.

What determines the $n$th order expansion coefficients of a given mode $\hat{\pi}$? In Fig. 2 we present scatter plots of $U^{(n)} \bullet \hat{\Psi}^{(n)}$ vs. $Ne$ where $e$ is the participation ratio defined in Eq. (1) above, and the order $n$ is varied between 3 to 8. Explicit expressions for expansion coefficients $U^{(n)} \bullet \hat{\pi}^{(n)}$ for particulate systems interacting via sphero-symmetric pairwise potentials are provided in Appendix C. We deliberately perform this analysis on normal modes $\hat{\Psi}$ in order to sample a larger range of degrees of localization compared to that seen for nonlinear modes. In Fig. 3 we demonstrate that the relation between the degree of localization of a mode and its associated expansion coefficients is general, and does not depend on whether harmonic or nonlinear modes are considered.

Our analysis clearly demonstrates the general trend that the more localized modes are, the higher the magnitude of their associated expansion coefficients. This means, in turn, that in addition to having low associated stiffnesses, solutions to Eq. (2) tend to be localized, which are precisely the two key characteristics of harmonic glassy modes [21].

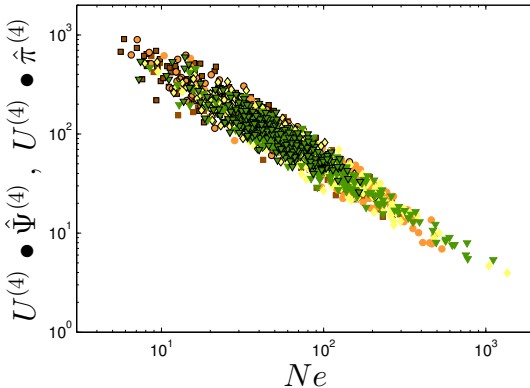

Figure 3: Expansion coefficients (see text for definition) vs. $N$ times the participation ratio $e$ of fourth order nonlinear glassy modes $\hat{\pi}$ (outlined symbols) and normal modes $\hat{\Psi}$ (symbols with no outline). Symbol shapes indicate the system size as described in the caption of Fig. 2.

## 5  Finding NGMs via an algebraic mapping

We proposed above that NGMs can be found by minimizing the appropriate cost function $\mathcal{G}_n(\hat{z})$ over directions $\hat{z}$. Here we spell out an iterative method for finding NGMs; we introduce the mapping

$$\mathcal{F}_n(\hat{z}) = \frac{\mathcal{M}^{-1} \cdot (U^{(n)} \bullet \hat{z}^{(n-1)})}{\sqrt{(U^{(n)} \bullet \hat{z}^{(n-1)}) \cdot \mathcal{M}^{-2} \cdot (U^{(n)} \bullet \hat{z}^{(n-1)})}} \,. \tag{8}$$

NGMs $\hat{\pi}$ of order $n$ are fixed points of the mapping $\mathcal{F}_n$, namely $\mathcal{F}_n(\hat{\pi}) = \hat{\pi}$. To see this, we rearrange Eq. (2) to read

$$\frac{U^{(n)} \bullet \hat{\pi}^{(n)}}{\mathcal{M} : \hat{\pi}\hat{\pi}} \hat{\pi} = \mathcal{M}^{-1} \cdot (U^{(n)} \bullet \hat{\pi}^{(n-1)}), \tag{9}$$

and we immediately find that

$$\frac{1}{\sqrt{(U^{(n)} \bullet \hat{\pi}^{(n-1)}) \cdot \mathcal{M}^{-2} \cdot (U^{(n)} \bullet \hat{\pi}^{(n-1)})}} = \frac{\mathcal{M} : \hat{\pi}\hat{\pi}}{U^{(n)} \bullet \hat{\pi}^{(n)}}, \tag{10}$$

which implies that $\mathcal{F}_n(\hat{\pi}) = \hat{\pi}$. We have verified numerically that the iterative process $\hat{z}_{q+1} = \mathcal{F}_n(\hat{z}_q)$ (where $q$ now indicates the iteration number) indeed converges to a solution $\hat{\pi}$ of Eq. (2), as demonstrated in a 3D system of $N = 2000$ in Fig 4 for $n = 4$, where the initial mode $\hat{z}_0$ was chosen to be random. We leave the detailed investigation of the convergence properties of the mapping $\mathcal{F}_n$ for future work.

## 6  Properties of nonlinear glassy modes

### 6.1  Spatial structure

In [21] it was reported that harmonic glassy modes are characterized by a disordered core, decorated by a field that decays at distances $r$ from their core as $r^{-2}$. In Fig. 5 we show the spatial decay profiles of the same low-frequency harmonic glassy mode studied in [21] in a system of $N = 10^6$ particles, in addition to the decay profiles of the nonlinear glassy modes iteratively mapped from the harmonic mode, as described above. We find that nonlinear modes of all orders decay as $r^{-2}$ as well. This result, together with the observation [11, 28] that in

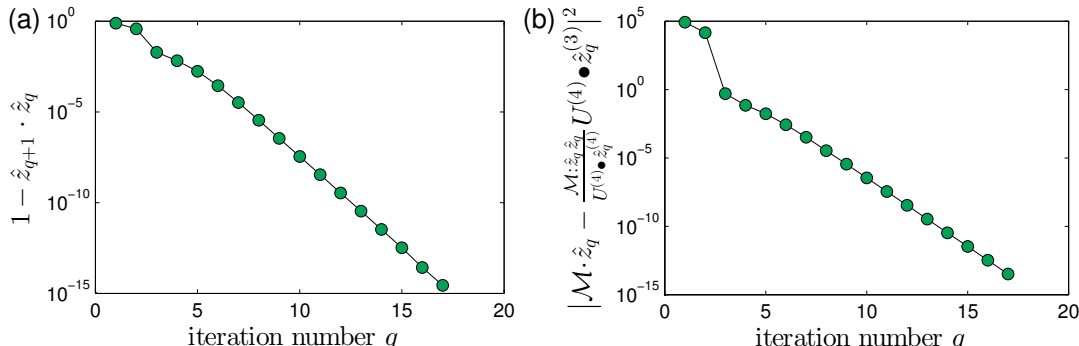

Figure 4: Proof of principle of the iterative mapping method introduced in this work. (a) Difference from unity of the overlap between the modes found in the $q$th and $q+1$th iterations, see text for further details. (b) Sum of squares of the difference between the RHS and LHS of Eq. (2), vs. iteration number. Notice the semilog axes scales.

2D systems and for $n=3$ nonlinear modes decay as $r^{-1}$, lead us to the assertion that all glassy modes, harmonic and nonlinear, decay as $r^{1-\bar{d}}$ away from their core.

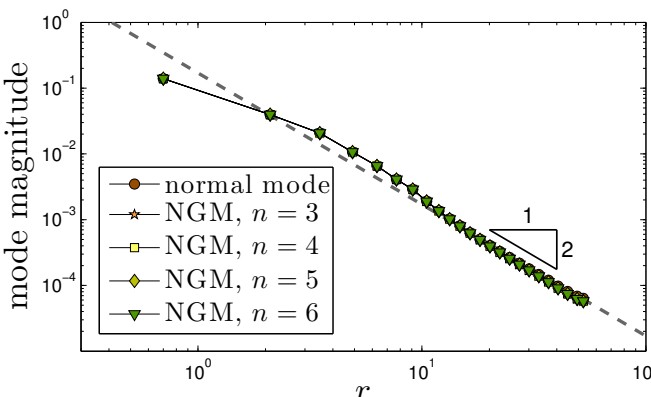

Figure 5: Spatial decay profiles (see [21,28] for a precise definition) of a harmonic glassy mode (circles) and of the higher order NGMs mapped from the harmonic mode, vs. the distance $r$ from the modes' core.

Eq. (2) suggests that NGMs can be thought of as the linear elastic response to a force $\propto U^{(n)} \bullet \hat{\pi}^{(n-1)}$ that depends on the mode $\hat{\pi}$ itself. Furthermore, the continuum-like decay of the far fields of NGM as described above resembles the linear elastic response to a localized force. To further establish this connection, we show in Fig. (6) the spatial decay profiles of the forces $U^{(n)} \bullet \hat{\Psi}^{(n-1)}$, calculated for the same low-frequency harmonic glassy mode analyzed in Fig. 5. Explicit expressions for $U^{(n)} \bullet \hat{\pi}^{(n-1)}$ for particulate systems interacting via sphero-symmetric pairwise potentials are provided in Appendix C. We find that the forces decay as $\sim r^{3-3n}$, as evident by the flattening of the spatial decay profiles of $r^{3n-3} |U^{(n)} \bullet \hat{\pi}^{(n-1)}|$ plotted in Fig. 6b. This scaling can be easily rationalized using similar arguments to those spelled out in [28] for the $n=3$ case. Fig. 6b also shows a strong signature of the core size of quasi-localized glassy modes, estimated in [21] at approximately 10 inter-particle distances in our computer glass. We have further checked that these results remain unchanged when the forces are calculating using NGMs of any order.

We conclude that highly localized forces are able to produce NGMs as the system's linear responses to those forces: for order $n = 4$, discussed in further detail in what follows, the

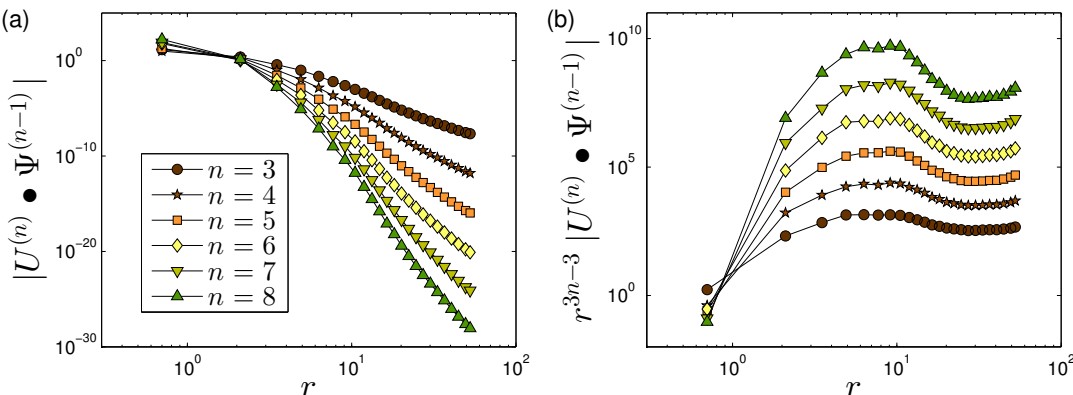

Figure 6: (a) Spatial decay profiles (see [21, 28] for a precise definition) of the forces $U^{(n)} \bullet \hat{\pi}^{(n-1)}$, vs. the distance $r$ from the modes' core. (b) Plotting the same decay profiles of (a), multiplied by $r^{3n-3}$, shows that $U^{(n)} \bullet \hat{\pi}^{(n-1)} \sim r^{3-3n}$ at large $r$.

force decays away from the core as $\sim r^{-9}$, i.e. extremely quickly. These results suggest that a close connection exists between the observed structural and energetic properties of various response functions to local perturbations in disordered elastic solids [29–32], and low-energy glassy modes. They further establish a link between the lengthscale that characterizes the crossover in such response functions to the scaling predicted by continuum elasticity [31], and the lengthscale that characterizes the core size of low-frequency glassy modes [21].

## 6.2 Energetics

We have established that NGMs share the same structural properties as harmonic glassy modes. We next turn to study the energy variations $\delta U(s)$ associated with displacing particles a distance $s$ along NGMs $\hat{\pi}$, i.e. following $\delta \vec{x} = s \hat{\pi}$. In Fig. 7 we show two examples of such variations calculated in systems of $N = 4000$ particles; panels (a),(b) show the energy variations for a low-frequency harmonic mode ($n = 2$), and for the third and fourth order NGMs calculated using the iterative method starting from the harmonic mode. Variations $\delta U(s)$ associated with higher order modes $n > 4$ are found to be similar in shape to the $n = 4$ case. We note that both cases presented in Fig. 7 are calculated from harmonic modes with very low vibrational frequencies $\omega / \omega_D \sim 10^{-2}$ where $\omega_D$ is the Debye frequency.

These examples demonstrate that low-energy excitations may appear with very different character in the glass; some correspond to 'double-well' excitations, whilst others are associated with simple, monotonic energy variations. We further find that only the third order modes are capable of robustly picking up 'double-well' type excitations. Another feature that stands out is the close similarity between the harmonic mode ($n = 2$) energy variations, and the $n = 4$ variations. We expand further on this point below.

In Fig. 7c,d we show the dependence on the order $n$ of the stiffness $\kappa \equiv \mathcal{M} : \hat{\pi}\hat{\pi}$ of the NGMs whose associated energy variations are displayed in panels (a),(b), respectively. Here and in the vast majority (99.7%) of cases studied, the 4th order modes (those that satisfy Eq. (4)) are found to have the *lowest stiffnesses* compared to 3rd *and* any higher order ($n > 4$) modes. We note that this analysis is performed on NGMs generated from the lowest harmonic mode of the system, and so stiffnesses associated with modes of any order $n > 2$ must be *larger* than the eigenvalue associated with the harmonic mode, which is the globally-minimal stiffness. The differences in stiffnesses associated to NGMs of different orders could be very small; for example, in the case displayed in Fig. 7a,c, the stiffness of the 8th order mode is approximately 2% larger than the stiffnesses of the 4th order mode. This is not always the case;

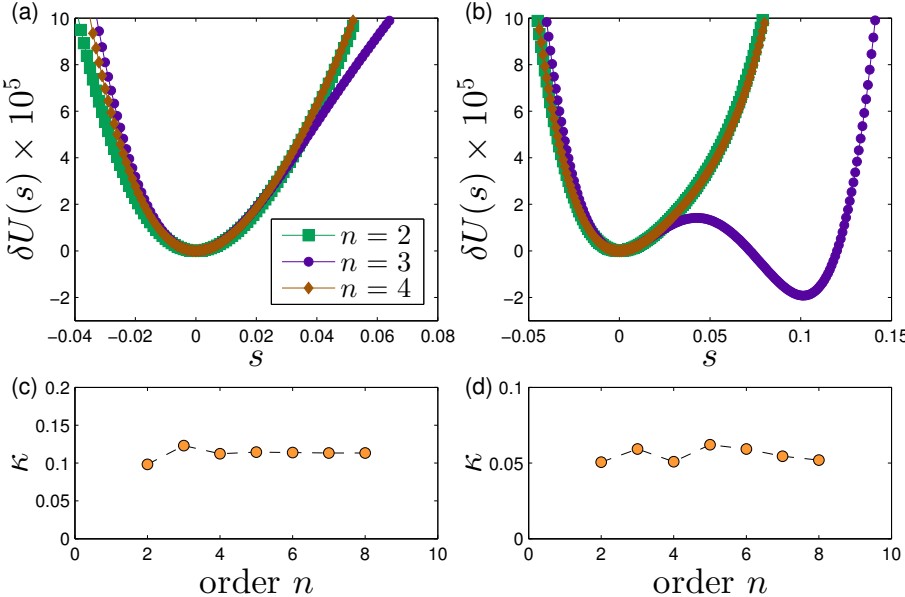

Figure 7: (a),(b) Energy variations $\delta U(s)$ obtained by displacing particles a distances $s$ according to $\delta \vec{x} = s \hat{\pi}$. (c),(d) Dependence of the stiffnesses of the NGMs on their order $n$, for the two cases displayed in panels (a),(b), respectively.

we also observe cases in which the 8th order mode is stiffer by a factor of two compared to the 4th order mode. We generally find that the tendency of higher order modes to have similar stiffnesses to those of 4th order modes increases as the frequency of the parent harmonic mode (from which the NGMs are calculated) decreases. We expand further on this trend in Sect. 8.

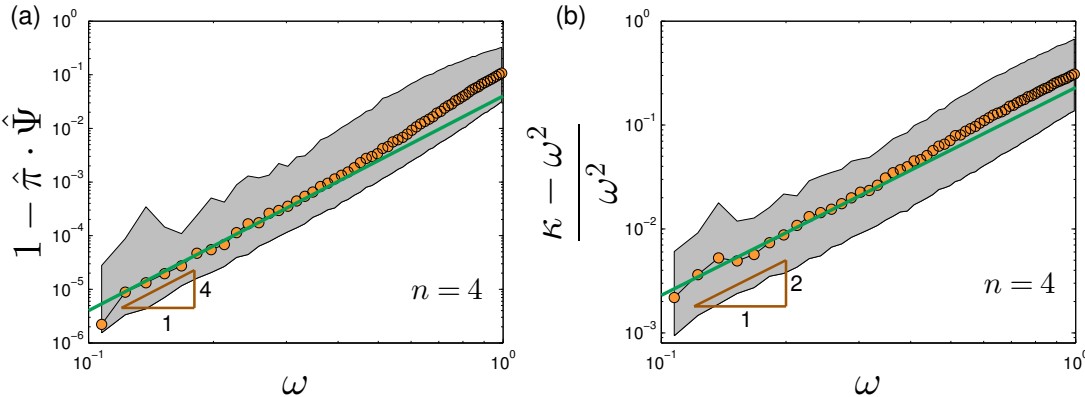

Figure 8: (a) Differences from unity of the overlap between HGMs and the 4th order NGMs mapped from them. The symbols represent the medians binned over frequency, and the shaded area represents the 2nd-9th deciles. (b) Relative increase in stiffnesses associated with the 4th order NGMs compared to those associated with the HGMs. Symbols represent the means binned over frequency, while the shaded areas are as in (a).

Having found that the fourth order modes are typically the softest amongst the entire family of NGMs, we turn now to a large-scale statistical study aiming at comparing the structure and stiffnesses associated with NGMs and those associated with harmonic glassy modes. We generated a large ensemble of over 500,000 glassy solid samples of $N = 2000$ particles, and for each of them we calculated the lowest frequency eigenmode $\hat{\Psi}$ of $\mathcal{M}$. In [21] it was shown that in glassy samples of our particular model, created under the same conditions, and having

the same number of particles, the lowest frequency mode is a (harmonic) *glassy* mode, i.e. its frequency is smaller than the frequency of the lowest Goldstone mode, and it is quasi-localized. We next used each of the harmonic modes calculated for each individual sample as the initial conditions for calculating the fourth order NGM using the iterative method described above.

Fig. 8 displays our first main result; in panel (a) we plot the difference from unity of the overlap between NGMs and their harmonic ancestors. The symbols represent the medians, binned over frequencies of the harmonic glassy modes, and the shaded areas cover the 2nd-9th deciles of each bin. We find that NGMs become gradually identical to their harmonic ancestors, as lower frequencies are considered. We find empirically

$$1 - \hat{\pi} \cdot \hat{\Psi} \sim \omega^4. \tag{11}$$

At this point we have no argument that explains this scaling.

In Fig. 8b we plot the relative increase in the stiffness associated with the calculated NGM on top of the stiffness associated with the harmonic modes (which is equal to their frequency squared) used to calculate the NGM, vs. the frequency of the harmonic mode. The symbols represents the means, binned over frequency, and shaded areas are as described for panel (a). We find that the stiffnesses $\kappa \to \omega^2$ as $\omega \to 0$, and moreover that the *relative* differences follow

$$(\kappa - \omega^2)/\omega^2 \sim \omega^2, \tag{12}$$

in consistency with the observation that $\hat{\pi} \to \hat{\Psi}$ as $\omega \to 0$.

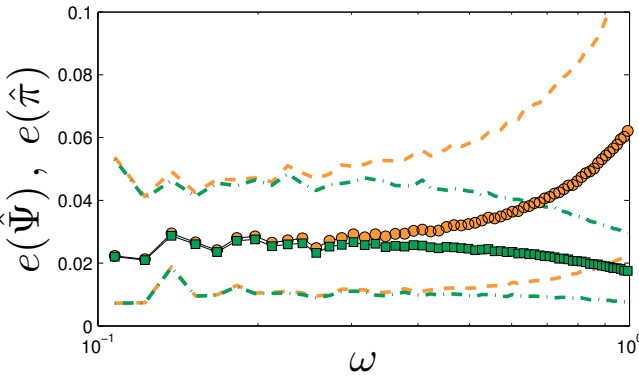

Figure 9: Comparison of participation ratios between harmonic modes ($e(\hat{\Psi})$, circles) and $n = 4$ NGMs mapped from them ($e(\hat{\pi})$, squares). Symbols represent means binned over frequency, and the dashed and dashed-dotted lines (pertaining to harmonic and nonlinear modes, respectively) mark the onset and offset of the 2nd and 9th deciles, respectively.

In Fig. 9 we compare the participation ratios of the harmonic modes (circles) with those of the NGMs generated from them (squares), see figure caption for further details. The two participation ratios converge at low frequencies, as expected from the converging overlaps as shown in Fig. 8a. Interestingly, at higher frequencies the two curves depart in opposite directions: the localization of harmonic modes begins to break down at higher frequencies (as seen also in Fig. 1b-d), whereas the localization of nonlinear modes becomes slightly more pronounced. This finding further supports that low-energy harmonic and nonlinear glassy modes are characterized by the same system- and preparation-protocol-dependent localization length.

We finally note that scaling laws describing the convergence of harmonic and nonlinear glassy modes at low frequencies is not universal for all orders $n$; in Appendix B we show that overlaps between third order NGMs and their harmonic ancestors converges to unity more

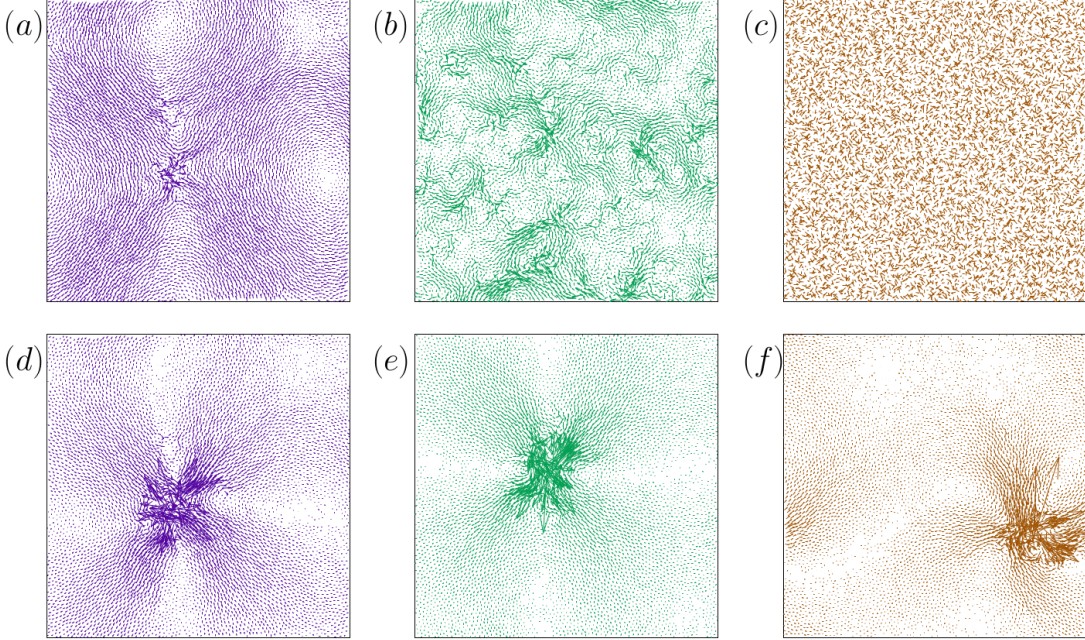

Figure 10: (a) A hybridized glassy and Goldstone vibrational mode. (b) The non-affine displacement responses to an imposed simple shear deformation (see e.g. [33] for definition). (c) A random mode. (d)-(f) display the fourth order nonlinear modes obtained using the iterative mapping method introduced in this work, with the modes of panels (a)-(c) used as the initial conditions, respectively.

slowly with frequency compared to 4th order NGMs. We further find that the increase in stiffnesses associated with 3rd order NGMs compared to the harmonic ancestors can be much more dramatic compared to 4th order NGMs, even at very low frequencies.

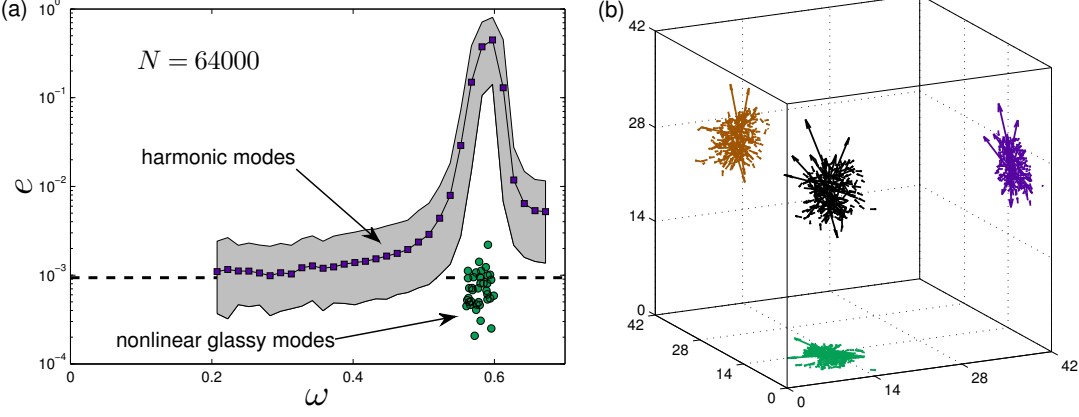

Figure 11: (a) Participation ratio of harmonic modes and of NGMs vs. frequency (see text for definitions), for systems of $N = 64000$. The squares and shaded area are the same as in Fig. 1d, and explained in that figure's caption. (b) Visualization of a NGMs found in the vicinity of the lowest Goldstone mode's frequency.

## 7 Disentangling of glassy and Goldstone modes

In this final section we show how our framework can be used to define and study glassy modes that cannot be realized as normal modes due to hybridizations with Goldstone modes. We first show in Fig. 10a-c examples of various delocalized modes in 2D (see caption for details), and the NGMs obtained by iteratively applying the mapping Eq. (8) with $n=4$ on these delocalized modes, until convergence to the modes displayed in panels (d)-(f). It is clear that NGMs exhibit the same structural features of harmonic glassy modes, and of the objects that destabilize under imposed deformations [11,34], regardless of the initial mode from which the NGMs are calculated.

In Fig. 11a we report our second main result; we replot the analysis of participation ratio data presented in Fig. 1d, calculated for low-frequency harmonic normal modes of 5000 samples with $N=64000$ particles. We superimposed on this plot a subset of data points (green circles) that represent the participation ratios of nonlinear glassy modes calculated in the same systems, starting from the nonaffine displacement response to a simple shear deformation (see e.g. [33] for definition; a 2D example is displayed in Fig. 10b) as the initial conditions for the calculation. We make this choice for computational convenience; similar NGMs in the same frequency range can be obtained using other choices for initial conditions, e.g. low frequency harmonic modes or linear displacement responses to localized forces.

We note that NGMs are not vibrational modes, and therefore are not, strictly speaking, characterized by vibrational frequencies. For the sake of comparison, we can nevertheless define an analog to vibrational modes' frequencies for a given NGM $\hat{\pi}$ as the square root of their associated stiffness, namely $\omega \equiv \sqrt{\kappa} = \sqrt{\mathcal{M} : \hat{\pi}\hat{\pi}}$ (recalling that the units of mass $m=1$). In Fig. 11a we present data points for NGMs with frequencies that fall precisely in the vicinity of the lowest Goldstone modes' frequency, where a clear void appears in the participation ratio data for harmomic modes. This shows that these same quasilocalized excitations that could be realized as harmonic normal modes in the absence of Goldstone modes, are accessible via a nonlinear glassy modes analysis. We further note that the calculated NGMs have smaller participation ratios relative to the harmonic modes, as also shown in Fig. 9; these are expected converge to the same values found for harmonic modes in the limit $\omega \rightarrow 0$, as indicated by the data of Fig. 9.

Fig. 11b displays one of these calculated NGMs, demonstrating again its quasilocalized nature; we only present components with magnitudes $\geq 10\%$ of the magnitude of the largest component of the mode.

## 8 Summary and Discussion

Advancing our understanding of the role of soft glassy excitations in the physics of structural glasses depends on our ability to robustly define and identify these excitations under any conditions, and in particular in large systems in which the low-frequency regime of the density of vibrational modes is overwhelmed by Goldstone modes. In this work we introduced a class of soft quasi-localized nonlinear excitations; these excitations are not normal modes of the harmonic approximation of the potential energy, and are therefore insensitive to the presence of Goldstone modes with comparable energies. We provide numerical evidence showing that these nonlinear excitations mimic very well harmonic glassy modes, both in their structural and energetic attributes.

Amongst the family of nonlinear excitations introduced in this work, of particular importance are the third and fourth order nonlinear modes, as defined in Eqs. (3) and (4), respectively. Third order ($n=3$) nonlinear modes have been previously coined *nonlinear plastic*

*modes* [11]; they were shown to be important for elasto-plastic processes, and to outperform harmonic modes in their ability to predict the loci and geometry of plastic activity in slowly sheared athermal glasses [28]. Here we further highlighted the sensitivity of $n=3$ modes to local 'double-well' like excitations, as seen in Fig. 7b. These excitations, which may be effectively detected and investigated using our proposed framework, could serve as the speculated two-level systems responsible for the anomalous thermodynamics of glasses at sub-Kelvin temperatures [1, 2].

Our key result regards however the utility of fourth order ($n=4$) modes, which were shown to be associated with very small energies, typically the smallest amongst all nonlinear modes of any other order $n \neq 4$. We assert that the $n=4$ modes are the best candidates to represent soft glassy excitations in glasses, in situations in which a conventional harmonic normal-mode analysis is futile.

Fourth order NGMs also constitute a robust definition of the low-energy excitations proposed in the Soft Potential Model framework [4, 12]. In this framework, a glass is envisioned to be partitioned into subsystems, and a soft quasi-localized mode is assumed to dwell in each such subsystem. The statistical properties of the expansion coefficients associated to those modes are discussed; the framework assumes that the 4th order expansion coefficients are frequency independent, as indeed verified in [21] for harmonic glassy modes, and therefore holds for NGMs as well, as implied by the data of Figs. 2 and 9.

In this work we did not touch extensively upon calculation issues of NGMs. Clearly, the practical usefulness of the framework introduced here depends on the future availability and robustness of techniques and algorithms for the detection of the entire *field* of low-energy NGMs. In Sect. 6 we do however discuss how NGMs can be thought of as the linear elastic response to a localized force field, with the general trend that higher order NGMs are given by responses to more localized force fields (see Fig. 6). A systematic study of the properties of those force fields whose linear elastic response pertain to NGMs is thus highly desirable – it will allow for the construction of smart initial conditions which will, in turn, be iteratively mapped onto the lowest-energy NGMs. The potential success of this suggestion is further reinforced by the observations that the structural and energetic properties of higher order NGMs do not depend strongly on their order $n$ (see e.g. Figs. 5, and 7c,d). This implies that linear responses to extremely localized forces, which are the simplest to construct, are likely to serve as useful heuristic ancentral modes that can be effectively mapped to the softest ($n=4$) NGMs. Future detection algorithms should be benchmarked against the computational approaches mentioned in the introduction [25, 26], in which auxiliary terms are added to the Hamiltonian with the goal of singling out soft glassy modes.

Another question that calls for further investigation regards the generality of our results; it would be interesting to test whether any of the qualitative results presented here depend on the properties of the model glasses studied, and if such dependencies are observed, how can they be explained.

## Acknowledgements

We thank Eran Bouchbinder, Gustavo Düring, and Yael Artzy-Randrup for fruitful discussions. We are grateful for computational facilities provided by the Chemical Physics Department of the Weizmann Institute of Science.

# A  Methods and Models

A detailed description of the model glass former used in this work, the definitions of microscopic units, and the preparation protocol with which glassy solids were created, can be found in [21]. In terms of the appropriate microscopic units, the particle density is set at $N/L^{d} = 0.82$, which sets the computer glass transition at $T_g \approx 0.5$. We find $\mu \approx 14$ for the athermal shear modulus, and the Debye frequency is $\omega_D \approx 17$. We used essentially the same model in 2D, except there the density was set to 0.86.

Vibrational modes were calculating using Matlab [35]. We used the iterative method explained in Sect. 5 for calculating nonlinear glassy modes. We deemed a nonlinear mode $\hat{\pi}$ converged if the ratio

$$\frac{|\mathcal{M} \cdot \hat{\pi} - \frac{\mathcal{M} : \hat{\pi} \hat{\pi}}{U^{(n)} \bullet \hat{\pi}^{(n)}} U^{(n)} \bullet \hat{\pi}^{(n-1)}|}{|\mathcal{M} \cdot \hat{\pi}|} < 10^{-10}. \tag{13}$$

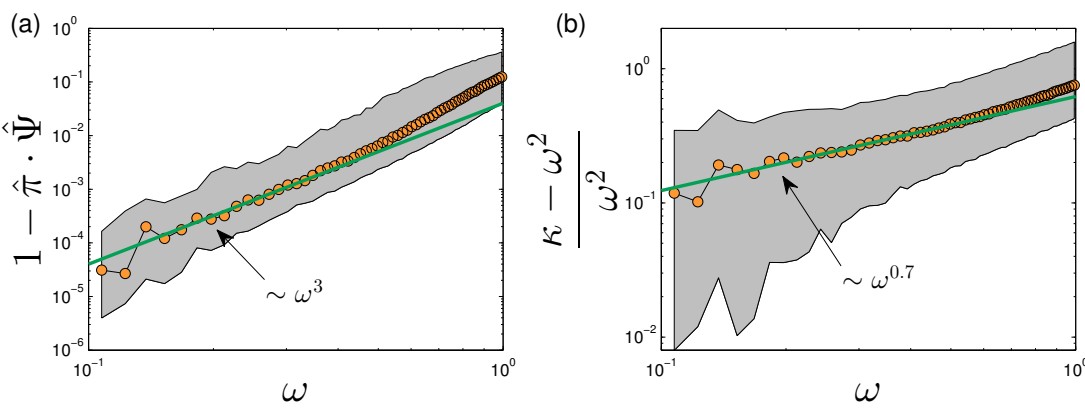

Figure 12: (a) Differences from unity of the overlap between HGMs and the 3rd order NGMs mapped from them. Symbols represent the medians, binned over frequency, while the shaded area cover the 2nd-9th deciles. (b) Relative increase in stiffnesses associated with 3rd order NGMs compared to those associated with the HGMs from which they were mapped. Symbols represent the means binned over frequency, while the shaded area is the same as for (a).

# B  Energies of $n = 3$ NGMs

In addition to calculating the fourth order NGMs of our ensemble of 500,000 glassy samples of $N = 2000$ particles, we also calculated the third order ($n = 3$) modes that solve Eq. (3) via the iterative method introduced in Sect. 5, starting from the lowest eigenmode of each system. In Fig. 12 we show the same analysis of $n=3$ NGMs' properties as presented in Fig. 8 for $n = 4$ NGMs. Interestingly, we find that the overlap of $n = 3$ NGMs and their harmonic ancestors also converges to unity as $\omega \to 0$, as the $n=4$ modes were shown to do in Fig. 8a, albeit more quickly. We further find that the stiffnesses of $n=3$ modes are significantly worse representatives of harmonic modes' stiffnesses, compared to $n=4$ modes.

## C Expressions for Taylor expansion coefficients and high-order forces

We consider systems in which the potential energy is given by a sum over pairwise interactions, of the form

$$U = \sum_{i<j} \varphi(r_{ij}), \tag{14}$$

where $r_{ij} \equiv |\vec{x}_{ij}|$ is the magnitude of the pairwise distance vector $\vec{x}_{ij} \equiv \vec{x}_j - \vec{x}_i$ between the $i$th and $j$th particles, and $\varphi(r)$ is a sphero-symmetric pairwise interaction potential. The full contraction of ($n$ instances of) the vector field $\vec{z}$, of $N \times d$ components, with the rank-$n$ tensor of derivatives of the potential energy $U^{(n)}$ reads

$$U^{(n)} \bullet \vec{z}^{(n)} = \sum_{i<j} \sum_{k=0}^{\lfloor n/2 \rfloor} \frac{\Phi_{n-k}(\vec{x}_{ij} \cdot \vec{z}_{ij})^{n-2k}(\vec{z}_{ij} \cdot \vec{z}_{ij})^k}{k!} \prod_{\ell=0}^{k-1} \binom{n-2\ell}{2}. \tag{15}$$

The products in the above equation and in what follows should be understood to be equal to unity if the upper bound index of the product is smaller then zero, e.g. for $k=0$ in Eq. (15). We next denote the $\ell$'th derivative of the pairwise potential $\phi(r)$ with respect to the pairwise distance $r$ as $\phi^{(\ell)}$, then $\Phi_\ell$ is spelled out for each interacting pair of particles as:

$$\Phi_\ell = \sum_{k=0}^{\ell-1} a_{k,\ell} \frac{\varphi^{(\ell-k)}}{r^{\ell+k}}, \tag{16}$$

for which the coefficients $a_{k,\ell}$ are given by the recursive relation

$$a_{k,\ell} = a_{k,\ell-1} - (\ell + k - 2)a_{k-1,\ell-1}, \tag{17}$$

with the boundary conditions

$$a_{0,1} = 1, \tag{18}$$
$$a_{k,\ell} = 0, \text{ if } k < 0 \text{ or } k \geq \ell. \tag{19}$$

We also work out the contraction of $U^{(n)}$ with one less vector $\vec{z}$ as

$$
\begin{aligned}
U^{(n)} \bullet \vec{z}^{(n-1)} = & \sum_{i<j} \left[ \sum_{k=0}^{\lfloor (n-1)/2 \rfloor} \frac{\Phi_{n-k}(\vec{x}_{ij} \cdot \vec{z}_{ij})^{n-2k-1}(\vec{z}_{ij} \cdot \vec{z}_{ij})^k}{k!} \prod_{\ell=0}^{k-1} \binom{n-2\ell-1}{2} \right] \vec{x}_{ij} \\
& + (n-1) \sum_{i<j} \left[ \sum_{k=1}^{\lfloor n/2 \rfloor} \frac{\Phi_{n-k}(\vec{x}_{ij} \cdot \vec{z}_{ij})^{n-2k}(\vec{z}_{ij} \cdot \vec{z}_{ij})^{k-1}}{(k-1)!} \prod_{\ell=0}^{k-2} \binom{n-2\ell-2}{2} \right] \vec{z}_{ij}.
\end{aligned}
\tag{20}
$$

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
