# Peer review of "Nonlinear modes disentangle glassy and Goldstone modes in structural glasses"

_SciPost Physics, doi:SciPost Phys. 1, 016 (2016)_

## Round 1 · Referee Report · Anonymous · 2016-12-16

Strengths
1- Studying a hot research topic
2- Proposing an extension of previous method to compute non-linear glassy modes
Weaknesses
1- Merit of the extension is not well demonstrated
Report
--Summary
Low frequency glassy modes originate from disordered structure of glasses thought to be related to thermodynamic, dynamic, and mechanical properties of glassy systems.
However, the glassy modes obtained by standard approaches such as the normal mode analysis are often hybridized with Goldstone modes.
Especially, at the thermodynamic limit, the standard approaches would completely fail to identify the low frequency glassy modes.
Thus, disentangling the glassy modes and Goldstone modes is an important research topic.
In the paper by Gartner and Lerner, the authors propose a numerical method to extract the low frequency non-linear glassy modes from disordered glassy configurations.
The non-linear glassy modes are obtained by minimization of the (n-th order) cost function which is designed so that solutions of the mode satisfy two characteristic features of known harmonic glassy modes, 1) low energy (small stiffness) and 2) localization (correlated with large expansion coefficients).
Whereas the third order (n=3) cost function introduced in the previous papers (Refs.[11,28]) corresponds to energy barrier between adjacent potential energy minima, there is no clear physical interpretation for higher order (n>3) cost functions.
Hence, the authors numerically check whether the obtained modes have same structural and energetic properties of the harmonic glassy modes.
As a result, the obtained modes, called non-linear glassy modes, share same spatial decay profiles and energy variations with the harmonic glassy modes, which justifies the proposed method.
By analyzing energetics of the non-linear glassy modes for n>2, the authors advocate that n=4 modes are the best candidates to represent soft glassy excitations in glasses.
--Comments
The motivation, problems, and proposed strategy, are well explained in the paper.
Also, numerical results are reasonable.
However, I have concerns about statement and technical issues described below.
a) n=4 modes are special?
In the paper, the authors assert that n=4 modes are the best candidates of soft glassy modes mainly because n=4 modes have the lowest stiffness among all non-linear modes with n>2.
However, it seems that this conclusion is not strongly supported by the presented data.
Because, according to FIG. 7(c) and (d), the stiffness as a function of n is nearly flat with taking similar values for n>3.
In addition to this, the authors demonstrated in Refs. [11,28] that n=3 modes also can predict soft region (quadrupole-like region in d=2) from hybridized modes.
From these observations, one would expect that structure of all non-linear modes of any n-th order are similar (not only statistical sense shown in FIG. 5) as long as they are obtained from a same given configuration.
If this expectation is true, the statement that n=4 modes are special might be weaken.
With this in mind, I would like to know data comparing structures of the mode with different orders from the same input, for example, by using the inner product of the modes as shown in FIG. 8(a).
b) Initial input modes for FIG. 11(a)
FIG. 11(a) shows disentangling the nonliner glassy modes and Goldstone modes.
In this plot, the authors employ the non-affine displacement response to shear deformation as the input for the iterative method.
Naively, one might think that starting from the harmonic modes (hybridized with Goldstone modes) is straightforward, because this way indeed fits the idea of disentangling the two modes.
Thus, it would be better to explain physical or technical reasons why the authors chose the non-affine displacement field.
Requested changes
1- Mistype in the caption of FIG. 1
before: the "horizontal" dash-dotted lines
after: the "vertical" dash-dotted lines
Author: Edan Lerner on 2016-12-21 [id 83]
(in reply to Report 1 on 2016-12-16)We thank the referee for carefully reading the manuscript, and for helping us improve its clarity. Please find below our responses to the referee’s comments, which are enclosed between horizontal lines.
* * *
a) $n\!=\!4$ modes are special?
In the paper, the authors assert that $n\!=\!4$ modes are the best candidates of soft glassy modes mainly because $n\!=\!4$ modes have the lowest stiffness among all non-linear modes with $n\!>\!2$. However, it seems that this conclusion is not strongly supported by the presented data. Because, according to FIG. 7(c) and (d), the stiffness as a function of $n$ is nearly flat with taking similar values for $n\!>\!3$.
* * *
We thank the referee for raising this interesting issue for discussion. Firstly, we would like to clarify that we have made the measurements and reported in the original manuscript that the stiffnesses of all $n\!>\!2$ nonlinear modes are, in more than 99% of the instances studied (a few thousands), larger than the 4th order modes’ stiffnesses. We certainly did not conclude that 4th order modes are the softest amongst all $n\!>\!2$ modes based on the data presented in Fig.7, which shows an analysis of stiffnesses for merely two instances of modes. Furthermore, notice that we did not overlook the flatness of the stiffness as a function of the order $n$ presented in Fig.7; instead, we explicitly commented on it in the original manuscript.
We believe that in a study focusing on low-frequency vibrational modes, singling out and focusing on the lowest-stiffness modes amongst the entire family of nonlinear modes is the most natural and relevant choice to be made. We wonder what other criteria aside from mode stiffness would be more relevant to our main goal, which was to find a micromechanical definition of soft glassy modes that are oblivious to the proximity of Goldstone modes with similar energies. Focusing on higher order nonlinear modes, which typically possess larger stiffnesses, seems sub-optimal with respect to our goal.
Nevertheless, an interesting question, albeit beside the main point of our work, concerns the statistical trends observed upon comparing stiffnesses of higher order modes to the stiffnesses of 4th order modes, beyond our observation that the 4th order modes are softer. In the revised manuscript we improved the discussion about the dependence of nonlinear modes’ stiffnesses on their order, and further mention that the tendency of higher order modes’ stiffnesses to be similar to 4th order modes' stiffnesses increases with decreasing frequency of their ancestral harmonic modes. We leave, however, the detailed statistical study of these trends for future work.
Finally, we elaborate further in what follows on why, in fact, the weak dependence of NGMs stiffness on their order $n$ can be seen as a strength of our approach.
* * *
In addition to this, the authors demonstrated in Refs. [11,28] that $n\!=\!3$ modes also can predict soft region (quadrupole-like region in $d\!=\!2$) from hybridized modes. From these observations, one would expect that structure of all non-linear modes of any $n$-th order are similar (not only statistical sense shown in FIG. 5) as long as they are obtained from a same given configuration.
* * *
There seems to be some confusion here: as described explicitly in the text, Fig.5 shows the decay profiles measured for individual NGMs calculated from the same ancestral harmonic mode. The decay profiles presented are not averaged over many realizations, and therefore the figure does not describe the similarity of different order NGMs in a statistical sense, but precisely in the sense mentioned by the referee.
* * *
If this expectation is true, the statement that $n\!=\!4$ modes are special might be weaken.
* * *
That $n\!>\!4$ modes can have similar (but almost always higher) stiffnesses compared to $n\!=\!4$ modes is beside the main point of our work, which is the demonstration that the stiffness and structure of (4th order) nonlinear modes converge to the stiffness and structure of the globally-minimal stiffness (harmonic) modes at low frequencies. Our observations clearly rule out the possibility that higher order modes show better convergence properties compared to $n\!=\!4$ modes.
Moreover, we see the similarity that $n\!=\!4$ modes bare to higher order modes as a potential strength of our approach, as discussed in the Summary and Discussion Section: NGMs can be thought of as the linear response to localized forces (see Section VI.a), with the general trend that higher order NGMs are given by responses to more localized forces. The similarity between high order NGMs to 4th order NGMs suggests that detecting the entire field of NGMs is possible by calculating the linear response to simple, localized forces, and using those linear responses as ancestral modes from which NGMs can be calculated. The similarity of 4th order NGMs to higher order NGMs increases the likelihood that such linear responses to localized forces serve as very good heuristics for obtaining useful ancestral modes. We expanded on this point in the Summary and Discussion Section of the revised manuscript.
* * *
With this in mind, I would like to know data comparing structures of the mode with different orders from the same input, for example, by using the inner product of the modes as shown in FIG. 8(a).
* * *
We reiterate that Fig. 5 precisely shows a comparison between the spatial structure of different order modes obtained from the same ancestor. A larger-scale statistical study of the differences between the structural and energetic properties of $n\!=\!4$ and $n\!\ne \!4$ modes, beyond the key observation that 4th order NGMs have the smallest stiffnesses amongst all other order NGMs, is outside of the scope of our work. The further investigation of the statistical differences between higher order NGMs, which are anyway more energetic and therefore of less interest, is left for future studies.
* * *
b) Initial input modes for FIG. 11(a)
FIG. 11(a) shows disentangling the nonlinear glassy modes and Goldstone modes. In this plot, the authors employ the non-affine displacement response to shear deformation as the input for the iterative method. Naively, one might think that starting from the harmonic modes (hybridized with Goldstone modes) is straightforward, because this way indeed fits the idea of disentangling the two modes. Thus, it would be better to explain physical or technical reasons why the authors chose the non-affine displacement field.
* * *
We thank the referee for pointing out this issue. We chose the non-affine displacement fields as ancestral modes for technical reasons: they are quickly computed, and are often mapped to low-energy nonlinear modes. There is nothing particular about this or any other choice for ancestral modes, as long as they can be mapped to the desired soft glassy modes. The important point is that NGMs can and do exist in the frequency regime where localized glassy modes cannot be represented by harmonic modes (due to strong hybridizations). We have clarified this point in the revised manuscript.

---

## Round 2 · List of Changes

1. We have fixed the mistake in the caption of Fig.1 (horizontal -> vertical).

2. We have expanded the discussion about the dependence of NGM stiffnesses on their order n.

3. We have expanded in the discussion section on how the mild dependence of NGMs’ structural and energetic properties on their order is, in fact, a strength of our approach and not a weakness, as it is expected to facilitate the construction of detection algorithms of the entire field of NGMs.

4. We have clarified our choice of using the non-affine displacement fields as ancestral modes for calculating NGMs with frequencies in the vicinity of the lowest Goldstone modes’ frequencies.

You are currently on this page

Resubmission 1610.03410v2 on 21 December 2016

---

## Editorial Decision

published